# Detection of Fluconazole Resistance in *Candida parapsilosis* Clinical Isolates with MALDI-TOF Analysis: A Proof-of-Concept Preliminary Study

**DOI:** 10.3390/jof12010009

**Published:** 2025-12-23

**Authors:** Iacopo Franconi, Benedetta Tuvo, Lorenzo Maltinti, Marco Falcone, Luis Mancera, Antonella Lupetti

**Affiliations:** 1Department of Translational Research and New Technologies in Medicine and Surgery, University of Pisa, 56126 Pisa, Italy; iacopo.franconi@phd.unipi.it (I.F.); benedetta.tuvo@ao-pisa.toscana.it (B.T.); l.maltinti3@studenti.unipi.it (L.M.); 2Mycology Unit, Pisa University Hospital, 56126 Pisa, Italy; 3Department of Clinical and Experimental Medicine, University of Pisa, 56126 Pisa, Italy; marco.falcone@unipi.it; 4Infectious Diseases, Pisa University Hospital, 56126 Pisa, Italy; 5Clover Bioanalytical Software, 18016 Granada, Spain; luis.mancera@cloverbiosoft.com

**Keywords:** MALDI TOF, azole resistance, *Candida parapsilosis*

## Abstract

In the context of evolving antifungal resistance and increasing reports of clinical outbreaks of non-albicans *Candida* spp. invasive infections, the rapid detection of resistant patterns is of the utmost importance. Currently, an azole-resistant *Candida parapsilosis* clinical outbreak is ongoing at Pisa University Hospital. Resistant isolates bear both Y132F and S862C amino acid substitutions. Based on the data and isolates retrieved during the clinical outbreak, mass spectrometry was used to investigate the differences between fluconazole-resistant and -susceptible clinical strains directly from yeast colonies isolated from agar culture media. A total of 39 isolates, 16 susceptible and 23 resistant, were included. Spectra were processed following a standardized pipeline. Several supervised machine learning classifiers such as Random Forest, Light Gradient Boosting Machine, and Support Vector Machine, with and without principal component analysis were implemented to discriminate resistant from susceptible isolates. Support Vector Machine with principal component analysis showed the highest sensitivity in detecting fluconazole resistance (100%). Despite these promising results, external prospective validation of the algorithm with a higher number of clinical isolates retrieved from multiple clinical centers is required.

## 1. Introduction

The landscape of fungal infections has completely changed over the last three decades, with a shift from AIDS-associated infections in the 1980s and 1990s to healthcare-related diseases from the early 2000s up to the present time [1]. Nowadays, major fungal threats are represented by *Candida* species and *Aspergillus* species with an evolving antifungal resistance pattern and are mainly correlated with immunocompromised hosts [2,3]. Multiple reports from all over the world have described the emergence of non-*albicans Candida* species in the form of clinical outbreaks, caused mainly by *Candida parapsilosis*, *Candida auris* and *Nakaseomyces glabratus*, with an associated increase in drug resistance rates [4,5].

The epidemiology and distribution of different *Candida* species causing invasive infections vary from one country to the other. Even if *C. albicans* remains the most frequently isolated pathogen all over the world, the prevalence and incidence of non-*albicans* species are increasing, as *C. parapsilosis* has emerged as the second most frequently isolated *Candida* spp. from the South European and Mediterranean countries, South Africa, some regions of China and Southeast Asia, and also Latin America [6,7,8,9,10,11,12]. North American, central and North European countries and central and East Asia have witnessed a rise in *N. glabratus* infections, as this species has become the second most frequently isolated *Candida* spp. from blood culture [13]. Italy has shown several reports regarding the dramatic rise in the percentages of *C. parapsilosis* isolated from blood cultures [14,15,16], especially in the clinical settings of ICUs and surgical wards. On top of that, another important species worthy of mentioning is *Candida auris*, which has caused multiple outbreaks worldwide and also in some regions of Italy [17,18]. This yeast has become the paradigm of multi-drug-resistant pathogens in the context of invasive fungal infections, as it has been associated with hand-to-hand spreading within healthcare facilities and the development of resistance against the three major classes of antifungals [19,20].

A clinical outbreak at Azienda Ospedaliero-Universitaria Pisana caused by azole-resistant *C. parapsilosis* has been previously described, and it is still ongoing [17]. In such clinical scenario a new mutation was found to be associated with fluconazole resistance, the C2585G mutation leading to the S862C substitution in the *CpMRR1* gene (*CPAR2_807270*) [17]. This gene encodes for the transcription factor of the efflux pumps CpCdr1b and CpMdr1b that are associated with expulsion of azole drugs from the yeast cell [21].

Currently, diagnosing fluconazole resistance is complex, requiring microbiological tests such as drug susceptibility testing and/or PCR methods. The gold standard to determine drug susceptibility in yeast is still a phenotypic test known as broth microdilution method that identifies Minimum Inhibitory Concentrations (MIC) on the growth of the microorganism for selected drugs according to the European Committee on Antimicrobial Susceptibility Testing (EUCAST) [22,23]. However, despite the accuracy, standardization and reliability of this diagnostic method, one disadvantage is the long reporting times, which are directly related to the growth time of the fungus, varying from one species to the other. For example, in the specific case of *C. parapsilosis*, the turnaround time is 48 h [24].

PCR methods, on the other hand, although faster, still require specific laboratory expertise, dedicated instruments and infrastructures [25,26]. Additionally, they are always dependent on a genomic material extraction step, inevitably increasing the turn-around time. Some PCR techniques might require isolated colonies of the microorganism and therefore the culturing of the fungal species, while others might be performed directly on biological samples [27]. Anyhow, PCR-based identification of species and antifungal resistant traits might still take hours to complete, and, even in the case of multiplex PCR, the spectrum of identifiable microorganisms and associated mutations depends on those included in the panel. Therefore, all PCRs include a selected number of pathogens and mutations involved in azole resistance [28]. Furthermore, confirmation of results with culture-based methods and broth microdilution test remains essential for the definition of drug resistance.

To date, routine identification at a species level of fungal microorganisms has been revolutionized by Matrix Associated Laser Desorption/Ionization Time of Flight (MALDI-TOF) Mass Spectrometry [29,30]. In the fungal field, MALDI-TOF has been successfully applied in both microorganism detection and identification for both yeasts and pluricellular fungi as well as investigation of antifungal susceptibility [31,32,33]. One way of assessing antifungal drug resistance in *C. albicans* was to evaluate changes in the spectral profile before and after exposure to serial dilutions of fluconazole. This method was named minimal profile change concentration (MPCC) and presented a correlation with a discrepancy of a maximum of two dilutions with the MIC values for fluconazole [34]. Specifically, the minimal drug concentration at which the cross-correlation with the spectra obtained from resistant isolates exposed to the maximal fluconazole concentration of 128 μg/mL (for the specific case of *C. albicans*) was higher than the result of the cross-correlation with the spectra of isolates not exposed to the drug was called MPCC [35]. Another way of assessing antifungal resistance with MALDI-TOF was to evaluate fungal growth via cellular density inferred from the surrogate data of peak intensity of a yeast suspension exposed to breakpoint levels of a specific drug [36].

All of the methods mentioned above still have the limit of adjunctive time required to perform the analysis. To date, no spectral analysis performed with MALDI TOF on *C. parapsilosis* has been able to distinguish between azole-susceptible and -resistant strains at the time when species identification is available. However, from a clinical perspective, having the possibility of detecting drug resistance at the time of species identification using a single yeast colony without any further processing of the yeast strain would definitely improve clinical practice. Detecting azole-resistant strains directly after culture isolation would allow us to choose the best antifungal regimen 48 h prior to the confirmatory antifungal susceptibility testing, and in the case of the same analysis performed directly on positive blood cultures even 72 h prior [37]. In addition, anticipating the detection of fluconazole resistance would also improve antifungal stewardship programs in the early definition of the appropriate drug regimen for the selected patient and prompt early application of infection control measures. 

In conclusion, to the best of our knowledge, no study has evaluated spectral differences using directly single yeast colonies between fluconazole-resistant and -susceptible *C. parapsilosis* strains bearing both Y132F and S862C substitutions. Within these diagnostic perspectives, the aim of this study was to evaluate the potential role of MALDI-TOF in the early diagnosis of azole-resistant *C. parapsilosis* strains isolated during the clinical outbreak at the Azienda Ospedaliero-Universitaria Pisana.

## 2. Materials and Methods

### 2.1. Fungal Isolates and MALDI-TOF Data Acquisition

A set of 16 susceptible and 23 resistant clinical isolates (n = 39) was collected at the University of Pisa. It comprised two batches. Batch 1 was formed by 20 isolates, including 10 susceptible and 10 resistant strains. Batch 2 comprised 19 isolates, including 6 susceptible and 13 resistant strains.

All fungal isolates were cultured on Sabouraud agar at 35 °C for 48 h. Single colonies were selected and subjected to protein extraction following the manufacturer’s recommendations. The yeast suspension was centrifuged and separated into 2 parts: the pellets and the supernatant. For the pellet part, the pellets were used for extraction and analysis of the protein spectrum. The pellets were extracted with the modified formic acid extraction following the Bruker Daltonics (Bremen, Germany) recommended protocol. Briefly, 300 µL of HPLC grade water and 700 µL of ethanol (Sigma-Aldrich, St. Louis, MO, USA) were added, mixed with vortex for 1 min, and centrifuged at 13,000 rpm for 3 min. The solution was discarded after centrifugation. Then, the pellets were dried; 70% formic acid (Sigma-Aldrich) was added until it covered the pellet. Acetonitrile (Sigma-Aldrich) was added to the equal volume of 70% formic acid (ratio 1:1) (20 µL each). The mixer was centrifuged at 13,000 rpm for 3 min. After centrifugation, one microliter of supernatant was spotted on the MALDI target plate (Bruker Daltonics) and dried at room temperature. One microliter of α-Cyano-4-hydroxycinnamic acid (HCCA) matrix was covered and dried at room temperature again. Then the sample was analyzed by MALDI-TOF MS both Bruker Microflex MALDI-TOF MS and Bruker Autoflex Speed MALDI-TOF MS. Each sample for MALDI-TOF MS was prepared in triplicates.

Mass spectra were acquired using a MALDI Biotyper (Bruker Daltonics, Bremen, Germany) operating under standard clinical conditions, using a mass range of 2k–20k Da. Spectra were collected in linear positive ion mode and exported in native format for subsequent preprocessing and analysis using the Clover MS Data Analysis Software (https://www.clovermsdataanalysis.com/, accessed on 3 March 2025, Clover Biosoft, Granada, Spain). All clinical isolates identified with MALDI-TOF were defined as *Candida parapsilosis* sensu stricto, as the current clinical software MBT HT Client IVD versione 5.2.330 licensed by Bruker Daltonics is able to differentiate within the *psilosis* complex (Bruker Daltonics, Bremen, Germany).

For each isolate, 3 biological replicates were performed in different days after 48 h of incubation and ≥48 technical replicates were obtained for each strain of batch 1 to account for the variability in sample preparation and instrument conditions. This resulted in the production of 1024 different spectra for batch 1 and 56 spectra for batch 2.

### 2.2. Antifungal Susceptibility Testing

All *C. parapsilosis* clinical isolates underwent evaluation with Merlin MICRONAUT-AM Antifungal Agents MIC© (Bruker Daltonics GmbH & Co. KG, Bremen, Germany) for the detection of antifungal susceptibility. The definitions of Minimum Inhibitory Concentrations, Clinical Breakpoints and results interpretation were performed according to the manufacturer’s instructions and the EUCAST reference document [38]. Culturing conditions were the same for isolation procedures.

### 2.3. Sanger Sequencing Analyses

Sanger sequencing analyses were performed on isolates from batch 1 for both *ERG11* (*CPAR2_303740*) and *CpMRR1* (*CPAR2_807270*) genes by Eurofins Genomic (Ebersberg, Germany). Unipro Ugene software (https://unipro.ru/; accessed on 3 April 2024, Unipro, Novosibirsk, Russia) [39] allowed for alignments of Sanger sequences. Primers were purchased from Merck (Merck KGaA, Darmstadt, Germany) using the online Oligonucleotide synthesis service (https://www.sigmaaldrich.com/IT/it/configurators/tube?product=standard, accessed on 30 July 2023). All 10 resistant isolates underwent Sanger sequencing. Isolates used in this study were as follows: CpA, CpB, CpE, CpD, CpF, CpG, CpH, CpI, CpN, CpO. Susceptible isolates sequenced were as follows: CpM, CpL, ATCC22019. Primers sequences and clinical isolates along with Sanger data and respective antifungal susceptibility testing profiles used in this study have been previously published [17].

### 2.4. Spectral Preprocessing

Spectra were processed following a standardized pipeline. Raw intensities were stabilized by square-root transformation, and high-frequency noise was reduced with a Savitzky–Golay smoothing filter (window length 11, polynomial order 3). Baseline effects were corrected using top-hat subtraction (factor 0.02). All spectra were subsequently averaged by replicates, aligned with a constant tolerance of ±2 Da and 600 ppm, normalized to the total ion current (TIC), and peak detection was performed with a relative intensity threshold of 0.05. Detected peaks were merged into a peak matrix using a 1 Da, 300 ppm tolerance window, where each value in the matrix represented the corresponding peak area for each spectrum.

### 2.5. Outlier Detection and Reproducibility Analysis

Potential outliers were assessed using a combination of principal component analysis (PCA) reconstruction errors and correlation-based analyses. The criterion for marking a spectrum as an outlier is that the distance from the mean in both analyses is greater than three times the standard deviation. No spectra met that criterion; therefore, all were retained. A reproducibility analysis was performed using the average spectra of each sample. Samples belonging to each group (susceptible and resistant) were compared with the averaged spectrum of all of its group. The D-index (differentiation index) is calculated by multiplying the difference of 1 to the correlation coefficient of each spectrum by 1000. Therefore, the D-index may adopt values between 0 and 2000. A value of 0 indicates identical spectra (or spectral ranges), a value of 1000 indicates completely non-correlated spectra, and a value of 2000 indicates completely negatively correlated spectra.

### 2.6. Machine Learning Training

For model training, the combined dataset was split into training and validation subsets using an 80/20 stratified partition. Several supervised machine learning classifiers were implemented to discriminate resistant from susceptible isolates. The algorithms included Random Forest (RF), Support Vector Machine with and without principal component analysis (SVM+PCA, SVM), Light Gradient Boosting Machine (LightGBM), Partial Least Squares Discriminant Analysis (PLS-DA), and k-nearest neighbors (KNN). All algorithms were trained with all the replicate spectra from the 20 samples in batch 1. Hyperparameters were optimized by exploratory comparison through all the possible value combinations. Final optimized values for batch 1 are presented in Table 1. Each algorithm was trained for each combination, and performance was compared by the accuracy obtained by a 10-fold cross-validation process, selecting the “Resistant” category as a positive. In 10-fold cross-validation, the initial dataset is randomly divided into 10 equally sized subsets, commonly known as “folds.” One of these subsets is set aside as validation data for model testing, while the remaining 9 subsets are used for training. This process is repeated 10 times, ensuring that each subset is used once as validation data. Finally, the 10 results are averaged to provide a single overall estimate. Because of the previous averaging of replicated spectra, cross-validation was grouped by isolate, as spectra from the same strain were not included in both training and testing.

Synthetic minority oversampling (SMOTE) was evaluated but excluded from final models, as better results were obtained without oversampling. Model performance was evaluated in terms of accuracy, balanced accuracy, sensitivity, specificity, and error rate.

### 2.7. Validation Within Batch 1

A validation step was performed using the validation datasets from batch 1. It was performed for the two algorithms providing the most promising results in the training step.

### 2.8. External Validation Using Batch 2

To assess the robustness of the classifiers trained in the first batch, the Random Forest (RF) and Support Vector Machine with PCA (SVM_PCA) models were directly validated using the isolates from batch 2. The raw spectra from these isolates were preprocessed with the same parameters applied during model training (variance stabilization, Savitzky–Golay smoothing, top-hat baseline subtraction, TIC normalization, alignment, and peak detection). Replicates corresponding to each isolate were averaged to generate representative spectra, which were then categorized as resistant or susceptible according to the clinical reference classification. These averaged spectra were submitted to the RF and SVM_PCA classifiers without retraining, and performance was evaluated by comparing predicted labels against the known categories.

### 2.9. Batch Comparison

After direct validation of batch 2 samples in batch 1-trained classifiers, a comparative analysis was performed to investigate potential differences between the two isolate batches. Outlier detection was carried out using PCA reconstruction error and correlation-based analyses, with thresholds set at three standard deviations from the mean. Spectra from both batches were analyzed jointly to identify isolates with systematic deviations from the overall dataset. These analyses confirmed differences between spectra in batch 2 vs. batch 1.

To mitigate the batch effect observed in direct validation, spectra from the first and second batches were combined into a single dataset for classifier development. Preprocessing was performed as previously reported. Replicates were averaged by isolate name to generate representative spectra, with manual curation applied to ensure inclusion of second-batch replicates. A combined peak matrix was made with all 51 samples, using the same steps and settings as for batch 1. The final dataset was split into training and validation sets at random, with an 80/20 split based on resistance category.

### 2.10. Machine Learning Models for the Combined Set and Model Evaluation for the Combined Set

Three supervised learning algorithms were prioritized for model training: Random Forest (RF), Light Gradient Boosting Machine (LightGBM), and Support Vector Machine (SVM), with and without principal component analysis (PCA). Partial Least Squares Discriminant Analysis (PLS-DA) and k-nearest neighbors (KNN) were additionally tested as comparators. Hyperparameters were optimized by 10-fold cross-validation to maximize balanced accuracy. Because all replicate groups were again averaged into a single spectrum, this cross-validation step never mixed spectra from the same isolates into training and test set. For RF, optimized settings included 200 estimators, maximum depth 30, minimum split size 5, and two samples per leaf. LightGBM was trained with 100 estimators, a learning rate of 0.1, and 16 leaves. SVM classifiers were trained using both linear and radial kernels; in the PCA variant, 229 principal components explaining 95% of the variance were retained. SMOTE was tested but excluded from final models, as optimal performance was achieved without it.

Performance was assessed in both cross-validation and independent validation sets. Metrics included overall accuracy, balanced accuracy, sensitivity, specificity, and error rate. Confusion matrices, receiver operating characteristic (ROC) curves, and precision–recall (PR) curves were generated for each classifier. Prediction models resulting from the best-performing configurations were exported for downstream use.

## 3. Results

### Reproducibility Analysis

Figure 1 and Figure 2 show Pearson’s coefficient (D-index) distribution box plot and violin plot for the reproducibility analysis of batch 1 spectra. These results indicate a clear correlation between spectra in each group, with two spectra per group having a lower correlation with the average but still within acceptable limits.

Table 2 shows the 10-fold cross-validation optimized training results for all ML algorithms using spectra from batch 1. The number of spectra in the training set adds up to 965, which corresponds to ~80% of the total number of spectra in the batch.

Training results indicate that RF has the best-balanced accuracy, being able to correctly classify 99.56% of the training set. It outperforms LGBM, which is a more complex algorithm. However, RF results could be suspiciously high, and that might indicate overfitting. The SVM models, both with and without PCA, appear to be promising; however, they permit a slightly higher training error, which often suggests a greater potential for generalization.

Table 3 shows the confusion matrix for the validation step by using the 20% remaining samples. Validation was applied only to those algorithms achieving a balanced accuracy in training superior to 99% (RF, SVM-PCA, SVM). These results confirm the overfitting bias of RF towards the positive category, showing a destitute generalization capability for susceptible isolates (48.96%). In contrast, both SVM with PCA and SVM without PCA were able to generalize correctly to the validation isolates, achieving a balanced validation accuracy of 97.69%, with a sensitivity of 100% (correctly identifying resistant isolates) and a specificity of 94.79% (correctly identifying susceptible isolates).

Table 4 shows the confusion matrix obtained when submitting the spectra from batch 2 to the selected classifiers. Direct validation of the first-phase classifiers using the second batch of isolates revealed poor generalizability. The Random Forest model (RF) maintained its bias towards the positive category (100%) but failed to identify any of the susceptible isolates (0/6, 0%), resulting in a balanced accuracy of 68.4%. Conversely, the SVM with PCA (SVM-PCA) achieved better recognition of susceptible isolates (5/6, 83.3%) but detected fewer than half of the resistant isolates (6/13, 46.2%), yielding a balanced accuracy of only 57.9%. The result of this model is consistent with the greater generalization power of SVM-PCA. However, these results demonstrated a strong batch effect and highlighted the limitations of models trained exclusively on the first dataset.

Following the unsatisfactory validation of the first-phase models, we performed an outlier analysis in the joined group formed by all spectra from both batches. This analysis identified eight samples as clear outliers (outlier index 5/5) and nine as potential outliers (index 4/5). These were all from batch 2. These results indicate a clear distinction between the samples from batches 1 and 2, which can explain the adverse validation results. Only one sample from the second batch is considered normal, compared to the first batch. Further analyses were performed to confirm these differences, like finding peaks whose presence/absence behaves as a strong discriminator of the spectrum belonging to either batch. Therefore, we concluded that direct validation of the first classifier with the samples of batch 2 is not satisfactory due to significant differences in the spectra compared to batch 1. The next step was to mix both batches for training and validation, as this could potentially improve the generalization of the final classifier for blind samples in routine. We formed a new training and validation set as described in the Material and Methods section, and we trained a battery of algorithms on the new combined set.

Random Forest: The RF model trained on the combined dataset achieved strong cross-validation metrics, with 96.14% training accuracy, 95.02% sensitivity, and 97.41% specificity. However, performance decreased in the independent validation set, where accuracy reached 82.7%. Specificity remained very high (97.9%), but sensitivity was modest (69.6%), indicating a tendency to misclassify resistant isolates as susceptible.

LightGBM: LGBM provided the most balanced results. Training accuracy reached 97.8%, with sensitivity and specificity both at 97.7%. Validation accuracy was 93.3%, with sensitivity of 91.1% and specificity of 95.8%. These results showed a strong trade-off between finding resistant isolates and not misclassifying susceptible ones.

Support Vector Machines: The SVM classifier achieved a training accuracy of 96.6% and, in the validation set, reached 91.4% accuracy. Sensitivity was maximized at 100%, ensuring that all resistant isolates were correctly identified, while specificity decreased to 81.3%. By contrast, the SVM without PCA provided a more balanced profile, achieving 96.6% training accuracy and 94.2% validation accuracy. Sensitivity remained high (98.2%), with an improved specificity of 89.9% compared to the PCA variant.

Other algorithms: PLS-DA and KNN underperformed relative to the other models, achieving validation accuracies of 70.2% and 39.4%, respectively. These classifiers were therefore not considered suitable for practical implementation.

Summary of classifier performance: a summary of training metrics across promising models is provided in Table 5. Among the tested algorithms, LGBM offered the most balanced performance between sensitivity and specificity, SVM+PCA maximized sensitivity by eliminating false-negative resistant predictions, and SVM without PCA provided a favorable compromise between sensitivity and specificity.

Table 6 shows that the validation of the classifiers trained on the integrated dataset confirmed substantial improvements in performance compared to the direct application of previous models. The RF classifier maintained high specificity but showed reduced sensitivity, leading to an overall validation accuracy of 82.7%. This procedure replicated the bias shown during the first training, discarding RF as a final candidate model.

LGBM provided the most balanced outcome, achieving a validation accuracy of 93.3%, with both sensitivity (91.1%) and specificity (95.8%) remaining high. The SVM with PCA reached 91.4% validation accuracy and maximized sensitivity (100%), ensuring all resistant isolates were correctly identified, although specificity dropped to 81.3%. This contrasts with the initial results, demonstrating that a more diverse dataset reduces the ability to generalize. By contrast, the SVM without PCA yielded the highest overall validation accuracy (94.2%) and a favorable compromise, with 98.2% sensitivity and 89.9% specificity. In comparison, PLS-DA and KNN performed poorly, with validation accuracies of 70.2% and 39.4%, respectively (not shown in Table 6). These results highlight the trade-offs between models: LGBM and SVM achieved the best balance, whereas SVM with PCA prioritized one category at the expense of the other.

## 4. Discussion

This study demonstrates that MALDI-TOF spectral data, when combined with machine learning, can effectively discriminate resistant from susceptible *C. parapsilosis* sensu stricto isolates bearing both Y132F and S862C substitutions directly from yeast colonies grown on solid medium with no further sample processing nor exposure to or incubation with antifungal drugs. Both mutations have been linked to azole resistance with the first being widely recovered from clinical resistant strains and the second recently reported in the ongoing clinical outbreak at Azienda Ospedaliero-Universitaria Pisana. To the best of our knowledge, this is the first study to ever evaluate spectral differences, directly from yeast colonies grown on solid medium, between fluconazole-resistant and -susceptible *C. parapsilosis* strains at MALDI TOF analysis.

Comparing these results with those previously reported in the literature, it seems reasonable to state that assessing susceptibility profiles at the time of microbial identification would improve clinical practice, reducing the delivery of inappropriate antifungal therapy, saving time for physicians and patients and lowering the consumption of unnecessary antifungal over-treatments, as for administration of echinocandins as first-line empiric therapy for candidemia. Previous studies highlighted the potential role of MALDI TOF in defining antifungal susceptibility tests, but that required exposure to serial concentrations of the target molecule and ended up requiring additional time after culture isolation of the yeast. The present work aimed at evaluating and highlighting traits on average spectra prior to exposure to the selected molecule, therefore identifying only the resistant phenotype to a specific molecule. This aspect might seem less informative than an antifungal susceptibility test, but it would be promptly available at the time of identification, saving up to 48 h, therefore delivering a valiant effort to steward antifungal empirical therapy. It must also be acknowledged that this implementation would only be considered as an adjunctive test in routine clinical practice rather than a surrogate for an antifungal susceptibility test. Plus, several limitations must be acknowledged. The initial training was conducted using the first batch of spectra. The work resulted in two models being proposed as candidates. However, direct validation against a second batch of isolates revealed that models trained on the initial dataset had poor generalizability, largely due to systematic batch effects. Outlier detection and biomarker analysis confirmed that the two batches displayed distinct spectral profiles, underlining the importance of dataset diversity in training robust classifiers. By integrating both batch 1 and 2 into a combined dataset and retraining the models, performance improved markedly. Among the algorithms evaluated, LGBM provided the most balanced classification, achieving high sensitivity and specificity in validation, and the SVM without PCA offered a favorable compromise between sensitivity and specificity. These trade-offs highlight that model selection should be guided by the clinical priorities of the diagnostic workflow: minimizing missed resistant infections versus limiting the misclassification of susceptible isolates. On the other hand, when focusing only of the classification of resistant strains, the SVM-PCA displayed the highest specificity. Although validation accuracy in this second phase did not always exceed that achieved during the initial analyses, the expanded dataset encompassed greater variability and therefore likely produced models with better generalization to real-world clinical settings. Still, it is of the outmost importance to state that results reported in current study are at a preliminary, proof-of-concept stage. We are tempted to speculate that clinical implications of the data presented in this study might help clinicians speed up results and ease the current clinical microbiology practice, giving information on resistant patterns directly from yeast cultures retrieved on solid media. However, it must be stressed once again that these results are based on a reduced number of isolates and require external validation from other cohorts and centers. Even if the number of spectra required to perform training and cross-validation of the algorithm is high (above 1000 spectra), they have been obtained by 39 clinical isolates in total; still statistical analyses have been performed but it appears clear that robustness of these results must be strengthened with adjunctive yeast strains and external validation. Lastly, another major study limitation that must be stressed is that models trained on the first batch of data showed a reduced performance on the second dataset. This odd occurrence was overcome only after retraining the model with combined data, achieving a satisfactory performance. The authors note that the differences between the series required the use of correction procedures; unfortunately, this means that the final model was partially fitted to the data. Therefore, authors acknowledge that, in future research, it would be advisable to repeat the model training using a single representative spectrum per isolate to validate that the classifier’s performance does not depend on the number of technical replicates.

## 5. Conclusions

These results indicate that machine learning applied to MALDI-TOF spectra is a promising strategy for antimicrobial resistance detection, provided that sufficient sample diversity is incorporated during model development. However, even if this model is a proof-of-concept at an early stage, it showed to be able to identify azole-resistant *C. parapsilosis* clinical isolates bearing the Y132F and S862C substitutions. Despite promising results and the potential advantages carried along with these results, the path towards the implementation of the algorithm in clinical practice requires more research efforts such as the expansion of the dataset and external validation. Future studies will require the application and use of the algorithm prioritizing the number of isolates tested rather than the number of spectra, in order to gain direct additional external validation with independent isolate cohorts from multiple clinical centers to confirm the reproducibility and clinical applicability of these models.

## Figures and Tables

**Figure 1 jof-12-00009-f001:**
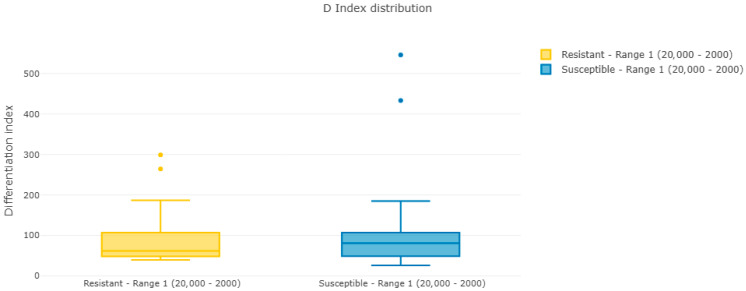
Box plot of Pearson’s coefficient distribution for samples in batch 1.

**Figure 2 jof-12-00009-f002:**
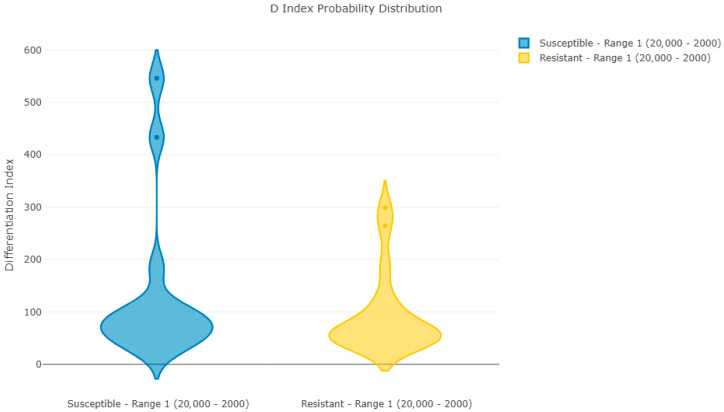
Violin plot of Pearson’s coefficient distribution for samples in batch 1.

**Table 1 jof-12-00009-t001:** Optimized values of all hyperparameters for final training with each ML model used in batch 1. Machine Learning models are highlighted in bold.

Random Forest	
Estimators	50
Max depth	10
Max features	184
Min samples per leaf	1
Min split size	1
**KNN**	
NCA	Applied
Number of neighbors	3
**PLS**	
Components	3
**SVM-PCA**	
PCA number of components	706 (95.02% variance)
C	1
**SVM**	
C	1
**LGBM**	
Number of estimators	100
Learning rate	0.01
Number of leaves	16
Minimum number of child samples	20

**Table 2 jof-12-00009-t002:** The 10-fold cross-validation optimized training results for several ML algorithms using the spectra from batch 1. Bold font indicates the best performance in the column.

Algorithm	No. Identified/Total No. (% Correct)		Balanced Accuracy (%)
	Susceptible (n = 395)	Resistant (n = 570)	
RF	**395/395 (100)**	565/570 (99.12)	**99.56**
LGBM	374/395 (94.68)	**567/570 (99.47)**	97.08
SVM	394/395 (99.75)	565/570 (99.12)	99.47
SVM-PCA	394/395 (99.75)	565/570 (99.12)	99.47
PLS-DA	384/395 (97.22)	561/570 (98.42)	97.82
KNN	373/395 (94.43)	551/570 (96.67)	95.55

**Table 3 jof-12-00009-t003:** Confusion matrix results for validation of trained ML algorithms using spectra from batch 1. Bold font indicates the best performance in its column.

Algorithm	No. Identified/Total No. (% Correct)		Balanced Accuracy (%)
	Susceptible (n = 96)	Resistant (n = 120)	
RF	47/96 (48.96)	113/120 (94.17)	74.07
SVM-PCA	**91/96 (94.79)**	**120/120 (100)**	**97.69**
SVM	**91/96 (94.79)**	**120/120 (100)**	**97.69**

**Table 4 jof-12-00009-t004:** Confusion matrix results for validation of trained ML algorithms using spectra from batch 2.

Algorithm	No. Identified/Total No. (% Correct)		Balanced Accuracy (%)
	Susceptible (n = 6)	Resistant (n = 13)	
RF	0/6 (0)	13/13 (100)	68.42
SVM-PCA	5/6 (83.33%)	6/13 (46.15)	57.89

**Table 5 jof-12-00009-t005:** A summary of training metrics for all ML models trained on the combined dataset. Bold font indicates the best result for the category.

Algorithm	No. Identified/Total No. (% Correct)		Balanced Accuracy (%)
	Susceptible (n = 193)	Resistant (n = 221)	
RF	188/193 (97.41)	210/221 (95.02)	96.14
LGBM	189/193 (97.93)	**216/221 (97.74)**	**97.83**
SVM-PCA	188/193 (97.41)	212/221 (95.93)	96.62
SVM	**190/193 (98.45)**	210/221 (95.02)	96.62

**Table 6 jof-12-00009-t006:** Validation performance of classifiers trained on the combined dataset. Best diagnostic performances are highlighted in bold.

Algorithm	No. Identified/Total No. (% Correct)		Balanced Accuracy (%)
	Susceptible (n = 48)	Resistant (n = 56)	
RF	**47/48 (97.92)**	39/56 (69.64)	69.64
LGBM	46/48 (95.83)	51/56 (91.07)	93.27
SVM-PCA	39/48 (81.25)	**56/56 (100)**	91.35
SVM	43/48 (89.58)	55/56 (98.21)	**94.23**

## Data Availability

All necessary data required to evaluate the conclusions of this study are reported in the tables in the main text.

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
