# Peer review of "Detection of Fluconazole Resistance in Candida parapsilosis Clinical Isolates with MALDI-TOF Analysis: A Proof-of-Concept Preliminary Study"

_jof, 2025, doi:10.3390/jof12010009_

Round 1

Reviewer 1 Report

  1. The manuscript uses the nomenclature Candida glabrata. However, the currently accepted name is Nakaseomyces glabratus.

  2. The study does not specify whether the isolates used belong to Candida parapsilosis sensu lato or C. parapsilosis sensu stricto. Since the study focuses on spectral profiles, this distinction is methodologically relevant. It is suggested that the identification of the isolates be confirmed by molecular biology techniques to rule out the presence of cryptic species within the C. parapsilosis complex. If this information already exists in a previous publication, it should be explicitly stated and properly cited.

  3. Although the model uses thousands of spectra, these are derived from only 39 clinical isolates. This can create a false sense of statistical robustness and lead to overfitting, especially considering the proposed clinical application. It is recommended that the manuscript explicitly clarify that the actual sample size corresponds to the number of isolates, not the number of spectra. Furthermore, it would be advisable to repeat the model training using a single representative spectrum per isolate to validate that the classifier's performance does not depend on the number of technical replicates.

The nomenclature Candida glabrata is used in the Introduction section. However, the currently accepted name is Nakaseomyces glabratus.

Author Response

Reviewer#1

We thank reviewer #1 for this useful and important observation that have certainly ameliorated the manuscript.

  1. The manuscript uses the nomenclature Candida glabrata. However, the currently accepted name is Nakaseomyces glabratus.

Corrected accordingly at line 41 and 48.

  1. The study does not specify whether the isolates used belong to Candida parapsilosis sensu lato or C. parapsilosis sensu stricto. Since the study focuses on spectral profiles, this distinction is methodologically relevant. It is suggested that the identification of the isolates be confirmed by molecular biology techniques to rule out the presence of cryptic species within the C. parapsilosis complex. If this information already exists in a previous publication, it should be explicitly stated and properly cited.

Corrected accordingly at Line 149. All clinical isolates identified with MALDI-TOF were defined a Candida parapsilosis sensu stricto, as current clinical software licensed by Burker diagnostics is able to differentiate within the psiosis complex (Bruker Daltonics, Bremen, Germany).

  1. Although the model uses thousands of spectra, these are derived from only 39 clinical isolates. This can create a false sense of statistical robustness and lead to overfitting, especially considering the proposed clinical application. It is recommended that the manuscript explicitly clarify that the actual sample size corresponds to the number of isolates, not the number of spectra. Furthermore, it would be advisable to repeat the model training using a single representative spectrum per isolate to validate that the classifier's performance does not depend on the number of technical replicates.

We would like to thank the Reviewer for outlining such important issues within this manuscript. Since some observations are overlapping with the suggestions provided by the second reviewer we decided to merge the answers within a new detailed paragraph in the discussion section aiming at elucidating all relevant issues and corrections raised by both reviewers.

Discussion

Line 413. Still, it is of the outmost importance to state that results reported in current study are at a preliminary, proof-of-concept stage. We are tempted to speculate that clinical implications of the data presented in this study might help clinicians fasten up results and ease current clinical microbiology practice giving information on resistant patterns directly from yeast cultures retrieved on solid media. However, it must be stressed once again that these results are based on few numbers of isolates and require external validation from other cohorts and centers. Even if the number of spectra required to perform training and cross-validation of the algorithm is high (above 1000 spectra) they have been obtained by 39 clinical isolates in total, still statistical analyses have been performed but it appears clear that robustness of these results must be strengthen with adjunctive yeast strains and external validation. Lastly, another major study limitation that must be stressed is that models trained on the first batch of data showed a reduced performance on the second dataset. This odd was overcome only after retraining of the model with combined data achieving a satisfactory performance. The authors note that the differences between the series required the use of correction procedures, unfortunately, this means that the final model was partially fitted to the data. The authors note that the differences between the series required the use of correction procedures, unfortunately, this means that the final model was partially fitted to the data.  Therefore, authors acknowledge that, in future research it would be advisable to repeat the model training using a single representative spectrum per isolate to validate that the classifier's performance does not depend on the number of technical replicates.

Detailed comments

The nomenclature Candida glabrata is used in the Introduction section. However, the currently accepted name is Nakaseomyces glabratus.

Corrected accordingly through the text.

Reviewer 2 Report

I suggest that the authors clearly state whether cross-validation was grouped by isolate – i.e., spectra from the same strain should not be included in both training and testing. Otherwise, the model may learn the ‘signature’ of a specific isolate instead of a general resistance pattern.

Models trained on the first batch of data performed poorly on the second dataset (different time points/measurement methods). Only after combining the data and retraining was satisfactory performance achieved. The authors rightly note that the differences between the series required the use of correction procedures. Unfortunately, this means that the final model was partially fitted to the data. I would ask the authors to discuss this limitation.

The title of the paper suggests the detection of strains with Y132F and S862C mutations using MALDI-TOF. MALDI-TOF does not directly identify genetic mutations.

I suggest changing the tone of the ‘Discussion’ and ‘Conclusions’ sections to be more cautious, clearly emphasising that the model presented is at the proof-of-concept stage and not a ready-made clinical tool. Currently, statements about improving clinical practice may be perceived as overly ambitious given the scale of the study and the lack of proper external validation.

Author Response

Reviewer#2

We thank reviewer # 2 for the important contribute given. We have managed to answer all points raised and re-assessed the tone and the clinical implications of the findings provided in this manuscript accordingly.

I suggest that the authors clearly state whether cross-validation was grouped by isolate – i.e., spectra from the same strain should not be included in both training and testing. Otherwise, the model may learn the ‘signature’ of a specific isolate instead of a general resistance pattern.

Line 214. Because of the previous averaging of replicated spectra, cross-validation was grouped by isolate as spectra from the same strain were not included in both training and testing.

Line 258. Because all replicate groups were again averaged into a single spectrum, this cross-validation step never mixed spectra from the same isolates into training and test set.

Models trained on the first batch of data performed poorly on the second dataset (different time points/measurement methods). Only after combining the data and retraining was satisfactory performance achieved. The authors rightly note that the differences between the series required the use of correction procedures. Unfortunately, this means that the final model was partially fitted to the data. I would ask the authors to discuss this limitation.

Discussion section

Line 420. Lastly, another major study limitation that must be stressed is that models trained on the first batch of data showed a reduced performance on the second dataset. This odd was overcome only after retraining of the model with combined data achieving a satisfactory performance. The authors note that the differences between the series required the use of correction procedures, unfortunately, this means that the final model was partially fitted to the data.

The title of the paper suggests the detection of strains with Y132F and S862C mutations using MALDI-TOF. MALDI-TOF does not directly identify genetic mutations.

The tile has been changed accordingly as follows:

Detection of Fluconazole-Resistance in Candida parapsilosis Clinical Isolates with MALDI-TOF Analysis: a proof-of-concept preliminary study

I suggest changing the tone of the ‘Discussion’ and ‘Conclusions’ sections to be more cautious, clearly emphasising that the model presented is at the proof-of-concept stage and not a ready-made clinical tool. Currently, statements about improving clinical practice may be perceived as overly ambitious given the scale of the study and the lack of proper external validation.

Discussion section

Line 413. Still, it is of the outmost importance to state that results reported in current study are at a preliminary, proof-of-concept stage. We are tempted to speculate that clinical implications of the data presented in this study might help clinicians fasten up results and ease current clinical microbiology practice giving information on resistant patterns directly from yeast cultures retrieved on solid media. However, it must be stressed once again that these results are based on few numbers of isolates and require external validation from other cohorts and centers. Even if the number of spectra required to perform training and cross-validation of the algorithm is high (above 1000 spectra) they have been obtained by 39 clinical isolates in total, still statistical analyses have been performed but it appears clear that robustness of these results must be strengthen with adjunctive yeast strains and external validation. Lastly, another major study limitation that must be stressed is that models trained on the first batch of data showed a reduced performance on the second dataset. This odd was overcome only after retraining of the model with combined data achieving a satisfactory performance. The authors note that the differences between the series required the use of correction procedures, unfortunately, this means that the final model was partially fitted to the data.

Line 428. Therefore, authors acknowledge that, in future research it would be advisable to repeat the model training using a single representative spectrum per isolate to validate that the classifier's performance does not depend on the number of technical replicates.     

Conclusions

Line 431. However, even if this model is a proof-of-concept at an early stage, it showed to be able to identify azole resistant C. parapsilosis clinical isolates bearing the Y132F and S862C substitutions. Despite promising results and the potential advantages carried along with these results, the path towards implementation of the algorithm in clinical practice requires more research efforts as expansion of the dataset, external validation. Future work will require application and use of the algorithm prioritizing the number of isolates tested rather than the number of spectra, in order to gain directly additional external validation with independent isolate cohorts from multiple clinical centers to confirm the reproducibility and clinical applicability of these models.

Round 2

Reviewer 2 Report

Dear Authors, 

thank you for taking my comments into account. At the same time, I would like to apologise for using bold type in my comments. I have only just noticed that this is how I sent my comments. It was not my intention.
I have no further comments.

None.